# Regulation of Cancer Stem Cells and Epithelial-Mesenchymal Transition by CTNNAL1 in Lung Cancer and Glioblastoma

**DOI:** 10.3390/biomedicines11051462

**Published:** 2023-05-17

**Authors:** Yeon-Jee Kahm, Uhee Jung, Rae-Kwon Kim

**Affiliations:** 1Department of Radiation Biology, Environmental Safety Technology Research Division, Korea Atomic Energy Research Institute, Yuseong-Gu, Daejeon 34057, Republic of Korea; 2Department of Radiation Life Science, University of Science and Technology, Yuseong-Gu, Daejeon 34057, Republic of Korea

**Keywords:** CTNNAL1, catenin alpha-like 1, cancer stem cells, epithelial-mesenchymal transition, CCL2

## Abstract

CTNNAL1 is a protein known to be involved in cell–cell adhesion and cell adhesion. Alterations in the expression or function of CTNNAL1 have been reported to contribute to the development and progression of various types of cancer. In breast cancer, CTNNAL1 has been reported as a cancer suppressor gene, and in melanoma and lung cancer, it has been reported as a cancer driver gene. However, due to a lack of research, its function remains unclear. In this study, it is shown that CTNNAL1 regulates cancer stem cells (CSCs) in lung cancer and glioblastoma and modulates their migration and invasion abilities. CSCs are known to play an important role in the malignant transformation of cancer. They have the ability to resist chemotherapeutic drugs and irradiation, which is a known obstacle to cancer treatment. We found that CTNNAL1 regulates the ability to resist irradiation. In addition, we observed that CTNNAL1 regulates the ability of cells to migrate and invade, a key feature of the epithelial to mesenchymal transition phenomenon associated with cancer metastasis. CTNNAL1 was also involved in the secretion of C–C motif chemokine ligand 2 (CCL2), one of the chemokines. CCL2 plays a role in the recruitment of immune cells to the tumor microenvironment, but in cancer, it is known to influence malignancy and metastasis. CTNNAL1 may be a novel target for treating lung CSCs and glioma stem cells and may be used as a marker of malignancy.

## 1. Introduction

CTNNAL1, also called catenin alpha-like 1, is a protein encoded by the human CTNNAL1 gene [1]. CTNNAL1 belongs to the alpha-catenin protein family and is involved in regulating cell adhesion and signal transduction [2]. CTNNAL1 is structurally similar to alpha-catenin and is a key component of adherens junctions, specialized structures that mediate cell-to-cell adhesion in epithelial tissues [3]. Like alpha-catenin, CTNNAL1 can participate in homologous interactions with cadherins in neighboring cells and bind to the cytoplasmic tail of cadherins, transmembrane proteins that mediate intercellular adhesion [4].

The epithelial-mesenchymal transition (EMT) involves cell-to-cell adhesI corfirmion [5]. Cells that undergo EMT have weakened cell-to-cell bonds and an increased ability to migrate and invade [6]. These abilities allow the cancer to metastasize. The EMT is a phenomenon that occurs early in cancer metastasis, and cancer cells that have undergone the EMT migrate from the primary cancerous tissue to the periphery of blood vessels, penetrating the blood vessel cell wall [7]. Cells that penetrate the blood vessel cell wall are not killed by physical stimuli within the vessel. This is a characteristic of cancer stem cells (CSCs), which have the ability to maintain a constant cell-cycle phase and resist external stimuli [8]. Cells traveling along blood vessels reach specific tissues, break through the vessel wall, and settle in other tissues [9]. Once settled, the cells form new cancerous tissue there, and CSCs are known to be able to self-renew and create new cancerous tissue [10]. CSCs are present in cancerous tissues in certain distributions. The cancers formed in each tissue can be categorized by specific marker proteins [11]. In general, CSCs are resistant to chemotherapy and irradiation, thus hindering cancer treatment [12,13]. 

Chemokines have been found to play an important role in helping cancer cells migrate along blood vessels and settle in specific tissues [14]. While chemokines are generally known to play an immune and anti-cancer role by recruiting immune cells [15], CSCs use chemokines as signaling stimuli to evade immunity or promote the malignancy of cancer [16]. CCL2 is one such chemokine and is well known for its role in recruiting immune cells; however, in many cancers, CCL2 is involved in cancer malignancy and is known to suppress the actions of immune cells [17,18]. 

## 2. Materials and Methods

### 2.1. Cell Culture

The human lung cancer cell lines used in the experiments were A-549 (human lung epithelial cells, ATCC ^®^ CCL-185™) and NCI-H460 (human lung epithelial cells, ATCC ^®^ HTB-177™). For the human glioblastoma cell line, U-87MG (Korea Cell Line Bank ^®^, Seoul, Republic of Korea) was used. The media were from Hyclone (Cytiva, Pittsburgh, PA, USA): RPMI 1640 MEDIUM (1X) (cat. no. SH30027.01) for the lung cancer cell line and DMEM/HIGHGLUCOSE (cat. no. SH30243.01) for the glioblastoma medium. Both of these media were supplemented with 10% FBS (cat. no. SH30919.03) and 1% penicillin–streptomycin solution (cat. no. SV30010) from the same company. Cells were cultured at 37 °C with 5% CO_2_ in a humidified incubator.

### 2.2. Sphere-Formation Assays

A sphere-formation assay was conducted by treating conditioned media (CM) into A-549 and U- 87MG cell lines. Conditioned medium was DMEM/F12 (1:1) (1X) from Invitrogen (Invitrogen; Thermo Fisher Scientific, Inc., Waltham, MA, USA) (cat. no. 11320-033), B27 serum-free supplement (cat. no. 17504-044), basic fibroblast growth factor (bFGF, cat. no. 13256-029), N-2 supplement (100X) (cat. no. 17502048), and epidermal growth factor from Sigma-Aldrich (cat. no. E9644, Burlington, CA, USA). 

### 2.3. Neutralization Assay

Antibody CCL2 (2 µg/mL; cat. no. MAB679; R&D SYSTEMS, Minneapolis, MN, USA) and normal mouse IgG1 antibody (1 µg/mL; cat. no. sc-3877; Santa Cruz Biotechnology, Inc., Dallas, TX, USA) were used in the neutralization assay. After antibody treatment, the antibody-treated A-549 cell line was incubated in a humidified incubator at 37 °C with 5% CO_2_ for 24–48 h.

### 2.4. Antibodies

The primary antibodies used for Western blot and immunocytochemistry were ALDH1A1 (cat. no. ab6192), ALDH1A3 (cat. no. ab129815), CD133 (cat. no. ab1998), and E-Cadherin (cat. no. ab15148) from Abcam (Cambridge, UK). Santa Cruz Biotechnology (Dallas, TX, USA) used β-actin (cat. no. sc-47778), SLUG (cat. no. sc-166476), SNAIL (cat. no. sc-10432), ZEB1 (cat. no. sc-25388), and Twist (cat. no. sc-15393). CCL2 (cat. no. MAB679) from R&D SYSTEMS (Minneapolis, MN, USA) and CD44 (cat. no. 3570), Nanog (cat. no. 4893), Oct-4 (cat. no. 2750), and Sox2 (cat. no. 3579) from Cell Signaling Technology (Danvers, MA, USA) were used. CTNNAL1 (cat. no. NBP1-33341; NOVUS Biologicals, Englewood, NJ, USA), N-Cadherin (cat. no. 610920; BD Transduction, Franklin Lakes, NJ, USA), and Vimentin (cat. MA5-14564; Invitrogen, Thermo Fisher Scientific, Waltham, MA, USA) were used.

### 2.5. Small Interfering RNA (siRNA) Mediated Knockdown of CTNNAL1

A siRNA treatment was used to knock down CTNNAL1 from the lung cancer cell line A-549 and the glioblastoma cell line U-87 MG. Cells were transfected with siRNA targeting CTNNAL1 (CAGACAAAACAGGAGUGAU, AUCACUCCUGUUUUGUCUG) and 10 pmol of Lipofectamine RNAi MAX reagent (cat. no. 13-778-150; Invitrogen; Thermo Fisher Scientific, Inc., Waltham, MA, USA). Stealth RNAi Negative Control Medium GC (cat. no. 12935-300; Invitrogen; Thermo Fisher Scientific, Inc., Waltham, MA, USA) was used as a negative control. After treatment, cells were incubated at 37 °C with 5% CO_2_ for at least 48 h.

### 2.6. Gene Expression Analysis

TRI reagent (cat. no. TR118; Molecular Research Center, Inc., Cincinnati, OH, USA) was used to extract RNA from A-549, NCI-H460, and U-87 MG cell lines. The amount of extracted RNA samples was measured by a spectrophotometer ASP-2680 (ACTGene, Piscataway, NJ, USA). A total amount of 1 µg of RNA samples was added to Maxime RT Premix (cat. no. 25082; iNtRON Biotechnology, Seongnam, Republic of Korea) to make cDNA. ALDH1A1 (F: TTAGCAGGCTGCATCAAAAC, R: GCACTGGTCCAAAAATCTCC, 34 cycles, 56 °C), ALDH1A3 (F: ACCTGGAGGGCTGTATTAGA, R: GGTTGAAGAACACTCCCTGA, 34 cycles, 57.5 °C), CD133 (F: CATGGCCCATCGCACT, R: TCTCAAAGTATCTGG, 34 cycles, 55 °C), CTNNAL1 (F: CTTAAGCTGGGTTTGCTCAC, R: CCCATCCGTTATTTTCCATCTG, 34 cycles, 57 °C) and GAPDH (F: AGTCAACGGATTTGGTCGTA, R: GTCATGAGTCCTTCCACGAT, 34 cycles, 56 °C) primers were used. PCR was performed using a T100 thermal cycler (Bio-Rad Laboratories, Inc., Hercules, CA, USA).

### 2.7. Western Blotting

The RIPA lysis buffer (cat. no. 20-188; Millipore, Burlington, CA, USA) with protease inhibitor cocktail tablets (cat. no. 1836153001; Roche, Basel, Switzerland) and phosphatase inhibitor cocktail tablets (cat. no. 04906837001; Roche, Basel, Switzerland) was used to lyse the cancer cell samples. The same amounts of proteins were separated by molecular weight on 8–12% SDS–polyacrylamide gels. After the separation of the proteins, they were transferred to Amersham™ Protran™ 0.2 µm NC (cat. no. 10600001; Amersham™; Cytiva, Pittsburgh, PA, USA). The transferred membranes were blocked by using phosphate-buffered saline (PBS) buffer containing nonfat milk (10%) for 30 min at room temperature. Membranes were probed with primary antibodies overnight in a 4 °C chamber. The next day, after washing membranes with Tris-buffered saline (TBS) buffer (cat. No. A0027; BIO BASIC, Markham, ON, Canada) containing Tween 20 (0.1%) (cat. No. TB0560; BIO BASIC, Markham, ON, Canada), HRP-linked secondary antibodies (anti-rabbit IgG; cat. No. 7074S or anti-mouse IgG; cat. No. 7076S; Cell Signaling Technology, Inc., Danvers, MA, USA) were treated for 2 h at room temperature. These membranes were visualized by using Western Blotting Luminol Reagent (cat. No. sc-2048; Santa Cruz Biotechnology, Inc., Dallas, TX, USA).

### 2.8. Immunocytochemistry

For the setting, 1 × 10^5^ cells were seeded on cover glass in 35 mm cell culture plates. On the next day, 4% paraformaldehyde (cat. no. P2031; Biosesang, Seongnam, Republic of Korea) was used for fixing. After 30 min fixation, the fixed cells were incubated with each primary antibody in Tris-buffered saline (TBS) buffer (cat. no. A0027; BIO BASIC, Markham, ON, Canada) containing Tween 20 (0.1%) (cat. no. TB0560; BIO BASIC, Markham, ON, Canada). ALDH1A1 (cat. no. ab6192; Abcam, Cambridge, UK), ALDH1A3 (cat. no. ab129815; Abcam, Cambridge, UK), CD44 (cat. 3570; Cell Signaling Technology, Inc., Danvers, MA, USA), CD133 (cat. no. ab1998; Abcam, Cambridge, UK), CTNNAL1 (cat. no. NBP1-33341; NOVUS Biologicals, Englewood, NJ, USA), E-cadherin (cat. no. ab15148; Abcam, Cambridge, UK), N-cadherin (cat. no. 610920; BD Transduction, Franklin Lakes, NJ, USA), and Vimentin (cat. no. MA5-14564; Invitrogen; Thermo Fisher Scientific, Inc., Waltham, MA, USA) were used as primary antibodies. The secondary antibodies used were Alexa Fluor™ 488 donkey anti-mouse IgG (H+L) (cat. no. A21202; Invitrogen; Thermo Fisher Scientific, Inc., Waltham, MA, USA) and Alexa Fluor™ 488 donkey anti-rabbit IgG (H+L) (cat. no. A21206; Invitrogen; Thermo Fisher Scientific, Inc., Waltham, MA, USA). The nuclei of cells were stained by DAPI solutions (cat. no. sc-24941; Santa Cruz Biotechnology, Inc., Dalls, TX, USA). Zeiss LSM510 meta-microscope (Carl Zeiss Microimaging GmbH, Göttingen, Germany) was used for imaging the stained samples.

### 2.9. Single Cell Assay

Ultra-low-adhesion 96-well plates (cat. no. 3474; Corning, Inc., Corning, NY, USA) were used in the single-cell assay. One or two siRNA-transfected A-549 cells were seeded in each well of 96-well plates, cultured with conditioned media (CM), and incubated (37 °C, 5% CO_2_) for 10–14 days. The inverted phase contrast microscope (magnification, ×400) was used to photograph the results.

### 2.10. Limited Dilution Assay

A limited-dilution assay was conducted using ultra-low-adhesion 96-well plates (cat. no. 3474; Corning, Inc., Corning, NY, USA) and conditioned media (CM). A total of 1, 10, 50, 100, 150, and 200 siRNA-transfected A-549 cells were plated in each well. After 10–14 days of incubation at 37 °C and 5% CO_2_ conditions, a phase contrast microscope (magnification, ×400) was used to photograph the results.

### 2.11. Invasion and Migration Assays

Invasion and migration assays were performed using uncoated chambers (cat. no. 3422, 8 μm pores; Corning, Inc., Corning, NY, USA). According to the protocol provided by the manufacturer, especially for the invasion assay, the chambers were coated with Matrigel^®^. The siRNA-transfected A-549 cell line was seeded in the upper chamber with 150 μL serum-free medium (Opti-MEM^®^, cat. no. 31985-070; Invitrogen; Thermo Fisher Scientific, Inc., Waltham, MA, USA) and for cell culture, the lower chamber was added with 800 μL RPMI 1640 MEDIUM (1X) (cat. no. SH30027.01; Hyclone; Cytiva, Pittsburgh, PA, USA) supplemented with 10% FBS. After incubation at 37 °C for 24 h in a humidified incubator with 5% CO_2,_ migrated and invaded cells from the upside to the bottom of the chamber were stained with crystal violet (cat. no. HT90132; Sigma-Aldrich; Merck KGaA; Burlington, CA, USA) for 2 min. Cells were counted using the phase contrast microscope (magnification, ×400).

### 2.12. Wound Healing Assay

A total of 2 × 10^5^ cells of siRNA-transfected A-549 were seeded in 35 mm cell culture plates (cat. no. 430165; Corning, Inc., Corning, NY, USA) and incubated (37 °C, 5% CO_2_) for 24 h to a density of 80%. A 200 μL pipette tip was used to create a wound on the monolayer of cultured cells by scratching. After the wound was created, floating cells caused by the scratch were washed out by PBS washing. After washing, RPMI 1640 MEDIUM (1X) (cat. no. SH30027.01; Hyclone; Cytiva, Pittsburgh, PA, USA) with 10% FBS (cat. no. SH30919.03; Hyclone; Cytiva, Pittsburgh, PA, USA) was added, and a phase contrast microscope (magnification, ×400) was used to picture the wound images. After 24 h of incubation (37 °C, 5% CO_2_), the wound images were used to compare the distance between the wound edges shown in the images.

### 2.13. Colony-Formation Assay and Irradiation

To analyze the colony-formation of siRNA-transfected A-549 cells, 1 × 10^3^ cells were seeded in 35 mm cell culture plates (cat. No. 430165; Corning, Inc., Corning, NY, USA) and incubated for 24 h at 37 °C, 5% CO_2_, in a humidified incubator. Cells were irradiated with 3 Gy γ-irradiation (Korea Atomic Energy Research Institute, Daejeon, Republic of Korea) and incubated in a humidified incubator (37 °C, 5% CO_2_) for 10–14 days after the irradiation. Crystal violet (cat. no. HT90132; Sigma-Aldrich; Merck KGaA; Burlington, CA, USA) was used to dye the colonies of the cells. The number of colonies was counted.

### 2.14. Cytokine Array

A cytokine array was conducted with the Human Cytokine Array Kit (cat. no. ARY005B; R&D SYSTEMS, Minneapolis, MN, USA). The day after transfection of A-549 cells with si-CTNNAL1, the cultured media was harvested for the cytokine array. The three-array membranes were incubated for 1 h in blocking buffer. During blocking the membranes, 15 μL of the reconstituted human cytokine-array detection antibody cocktail was added to the prepared samples (cultured media) and incubated for 1 h at room temperature. After 1 h, blocking buffer was removed, and prepared samples within the human cytokine array detection antibody cocktail were added. The day after the incubation at 4 °C, each membrane was washed a total of three times and then visualized by a chemical reagent mixture.

### 2.15. Statistical Analysis 

All experiments were performed in at least three independent experimental replicates, and results are presented as the mean ± standard deviation. The exact n value for each is indicated in the corresponding figure legend. All graphs were validated by comparing data using a two-sided paired Student’s *t*-test. Statistically, *p* < 0.05 was considered to indicate a significant difference.

## 3. Results

### 3.1. Identification of CTNNAL1 as a Potential Target in ALDH1^+^ Lung Cancer Cells 

In a previous study, we identified genes that are overexpressed in ALDH1^+^ lung cancer cells [19]. To find new targets, we compared these genes to a list of genes that are overexpressed in CD133^+^ GBM. The CD133^+^ GBM gene results we used were obtained from another group [20]. From the comparison, we identified the CTNNAL1 gene and conducted experiments to determine its function. According to previous studies, A549 cells are composed of more than 80% ALDH1^+^ cells [19], while only about 2% of H460 cells express ALDH1^+^ [21]. Therefore, we compared the expression levels of CTNNAL1, ALDH1, and CD133 by Western blot (WB) using A549 and H460 cells (Figure 1A). The results showed that A549 cells had higher expression of CTNNAL1 compared to H460 cells, as well as higher expression of CSC marker proteins ALDH1 and CD133. CD133 is a marker protein of glioma stem cells (GSCs) and a common marker protein of lung CSCs, so we used it in the subsequent experiment. H460 with low expression of ALDH1 was experimented with under conditioned media (CM) culture conditions (Appendix A). This resulted in increased expression of CSC marker proteins, including CTNNAL1. To compare the expression at the gene level, we performed PCR experiments (Figure 1B). The results showed that at the gene level, A549 cells had higher expression of ALDH1 and CD133, including CTNNAL1, than H460 cells, as shown in Figure 1A. To verify the experimental results, we used the immunocytochemistry (ICC) method (Figure 1C). The expression of ALDH1, CD133, and CTNNAL1 was, respectively, higher in A549 cells than in H460 cells.

### 3.2. CTNNAL1 Gene Silencing Reduces Cancer Stem Cell Characteristics and Enhances Sensitivity to Irradiation in Lung Cancer

To determine whether CTNNAL1 affects CSCs, we performed sphere formation experiments (Figure 2A). The results showed that the formation of spheres in the si-CTNNAL1 group was significantly lower, and their size was smaller. A major characteristic of CSCs is their ability to self-renew. To measure this self-renewal ability, a single-cell assay was performed (Figure 2B). The results revealed that when the CTNNAL1 gene was silenced using siRNA, the self-renewal capacity was significantly reduced. In addition, a limited-dilution assay was performed to compare the difference in self-renewal capacity, and it was found that the sphere formation capacity was reduced in the si-CTNNAL1 group (Figure 2C). To determine the changes in the marker proteins of lung CSCs, the expression levels of the proteins were compared after treatment with siRNA (Figure 2D). The results showed that CSC marker proteins CD133, ALDH1A1, ALDH1A3, and CD44 were reduced when the CTNNAL1 gene was suppressed. When H460 cultures cultured in CM conditions were treated with siRNA to suppress the expression of CTNNAL1, CSC marker proteins were reduced (Appendix A). Figure 2E shows a comparative analysis of the expression levels of SOX2, Oct-4, and Nanog, which are known CSC regulatory proteins, and all of their levels were reduced when CTNNAL1 expression was suppressed by siRNA. ICC was performed to visually analyze the CSC marker proteins regulated by CTNNAL1 (Figure 2F). The results showed that, similar to the results in Figure 2D, the expression of CSC marker proteins was reduced in the cells of the si-CTNNAL1 knockdown group. CSCs have the ability to resist irradiation. To determine whether CTNNAL1 is involved in the ability of CSCs to resist irradiation, a colony-forming assay was performed (Figure 2G). The results revealed that colony formation was dramatically reduced in the group where CTNNAL1 expression was suppressed by siRNA. 

### 3.3. CTNNAL1 Regulates EMT Phenomena and Cell Motility

CSCs are known to possess features of the EMT [22,23]. Given this, we identified changes in EMT marker proteins by CTNNAL1 (Figure 3A). The epithelial marker E-cadherin was increased when CTNNAL1 was inhibited by siRNA, and the mesenchymal markers N-cadherin and Vimentin were decreased. When H460 cultures cultured in CM conditions were treated with siRNA to suppress the expression of CTNNAL1, EMT marker proteins were regulated (Appendix A). In addition, overexpression of CTNNAL1 in H460 cells decreased the epithelial marker E-cadherin and increased the mesenchymal markers N-cadherin and Vimentin (Appendix A). EMT regulatory proteins Snail, Slug, TWIST, and ZEB1 all tended to be reduced when si-CTNNAL1 was treated (Figure 3B). As presented in Figure 3C, ICC experiments were performed to observe changes in EMT marker proteins. Similar to the results in Figure 3A, the expression of E-cadherin was increased and the expression of N-cadherin and Vimentin, respectively, decreased in cells in which CTNNAL1 expression was suppressed by siRNA. The EMT is closely related to cell motility. Therefore, a wound-healing assay was performed to measure the migratory ability of cells (Figure 3D). The results showed that the migratory ability of cells in the si-CTNNAL1-treated group was significantly reduced. In addition, the migratory and invasive abilities of cells were observed using a Boyden chamber, and the migratory and invasive abilities of cells were reduced in the group in which the expression of CTNNAL1 was suppressed by siRNA (Figure 3E). Overexpression of CTNNAL1 in H460 cells increased the migration and invasion capacity of the cells (Appendix A).

### 3.4. CTNNAL1 Regulates CCL2 to Affect Cancer Stem Cells and EMT

Cancer cells are known to rely on cytokines to drive malignancy and regulate the behavior of immune cells. Therefore, we performed a cytokine array to identify cytokines regulated by CTNNAL1 (Figure 4A). The results showed that CCL2 was significantly reduced in cell lines treated with si-CTNNAL1. To determine if CTNNAL1 regulates CCL2, we performed WB (Figure 4B). In cells where CTNNAL1 was reduced by siRNA, CCL2 was reduced. To determine whether there is a mutual regulation between CTNNAL1 and CCL2, WB was performed after treatment with a CCL2 neutralizing antibody (Figure 4C). Inhibiting CCL2 signaling did not change the protein expression of CTNNAL1; however, the expression of CCL2 was reduced by the CCL2 neutralizing antibody. To determine whether secreted CCL2 affects CSCs, we checked the CSC marker proteins CD44, ALDH1A1, and ALDH1A3 (Figure 4D). The results showed that the expression of all marker proteins was reduced. Furthermore, we observed EMT marker proteins after treatment with CCL2 neutralizing antibodies and found that E-cadherin was increased by CCL2, while N-cadherin and Vimentin were decreased (Figure 4E).

### 3.5. CTNNAL1 Regulates Stemness and EMT in GBM Cells and Controls CCL2 Secretion

We identified CTNNAL1 as a gene that is commonly overexpressed in ALDH1^+^ lung cancer cells and CD133^+^ GBM cells; however, its function in GBM cells is unknown. To determine the role of CTNNAL1 in GSCs, we conducted the following experiments. We knocked down the expression of CTNNAL1 using siRNA in U87 cells and then treated them with CM to form spheroids to check the expression of GSC marker proteins (Figure 5A). When U373 cells cultured in CM conditions were treated with siRNA to suppress the expression of CTNNAL1, CSC marker proteins were regulated (Appendix A). In cells with suppressed expression of CTNNAL1, we found that the expression of the marker proteins ALDH1A1, ALDH1A3, and CD133 was suppressed. To visually analyze this, we performed ICC experiments (Figure 5B). si-CTNNAL1 treated cells had significantly reduced expression of ALDH1A1 and CD133. To compare the degree of sphericity, a hallmark of CSCs, we performed a spheroidization assay (Figure 5C). In the group that silenced CTNNAL1 with siRNA, not only the size of the spheres but also their number were significantly reduced. In GBM cells, we observed whether CTNNAL1 is involved in EMT phenomena. We treated U87 cells with siRNA to suppress the expression of CTNNAL1 and harvested cells after CM treatment to induce spheroidization. We examined the expression of EMT marker proteins E-cadherin, N-cadherin, and Vimentin and found that the expression of epithelial marker E-cadherin increased and the expression of mesenchymal markers N-cadherin and Vimentin decreased in cells with suppressed CTNNAL1 expression (Figure 5D). When U373 cells cultured in CM conditions were treated with siRNA to suppress the expression of CTNNAL1, EMT marker proteins were regulated (Appendix A). ICC experiments were performed to visually verify the changes in EMT marker proteins (Figure 5E). The results showed that the expression of E-cadherin was increased and the expression of N-cadherin and Vimentin, respectively, decreased in cells whose expression of CTNNAL1 was suppressed by siRNA treatment, as shown in Figure 5D. To determine the ability of cells to migrate and invade, which are the most important features of the EMT, experiments were performed using a Boyden chamber (Figure 5F). The migration and invasion of cells in the CTNNAL1 siRNA knockdown group were significantly reduced. The secretion of CCL2 was regulated by CTNNAL1 in lung cancer cells. To confirm the secretion of CCL2 by CTNNAL1 in brain CSCs, the expression of CCL2 was observed after inhibiting CTNNAL1 using siRNA (Figure 5G). The results showed that CCL2 was regulated by CTNNAL1.

## 4. Discussion

Of all deaths from cancer worldwide, 18.4% are lung cancer patients with a poor prognosis [24]. For gliomas, the chance of surviving more than 10 years after diagnosis is less than 1% [25], and patients with GBM usually die within 12 to 18 months of diagnosis [26]. CSCs are known to play an important role in tumor initiation, progression, and resistance to treatment [27]. ALDH1 has been identified as a marker of lung CSCs and targeting ALDH1^+^ cells may be a potential strategy for the treatment of lung cancer [28]. In a previous study, we identified genes that are overexpressed in ALDH1^+^ lung cancer cells [19]. To find new targets, we compared these genes to a list of genes that are overexpressed in CD133^+^ GBM [20] and thereupon found the CTNNAL1 gene.

CTNNAL1 plays an important role in cell–cell adhesion and signal transduction [3]. Previous studies have shown that CTNNAL1 is overexpressed in various types of cancer, including lung cancer, and is associated with tumor progression and a poor prognosis [29]. Silencing the CTNNAL1 gene reduces CSC properties and enhances sensitivity to irradiation in lung cancer.

CTNNAL1 has also been shown to regulate EMT events and cell motility. The same results were obtained in GBM cells and lung cancer cells, suggesting that CTNNAL1 plays the same role in both cancer types. In particular, CTNNAL1 is a protein involved in cell adhesion, which may have a direct effect on the EMT. The signaling mechanisms that are involved will be investigated further.

The present study also showed that CTNNAL1 affects CSCs and EMT by regulating CCL2. In the process, the expression of CCL2 was inhibited by a CCL2 neutralizing antibody. In some papers [17,30], CCL2 has been reported to be associated with an autocrine loop. It is believed that the autocrine loop is mediated by CCL2–CCR2 signaling, but the detailed signaling mechanism is not well understood. The autocrine loop of CCL2 will be investigated in the next study. CCL2 is known as one of the chemokines that recruit inflammatory cells. However, its action in cancer cells is involved in the malignant transformation of cancer and suppresses immunity. Therefore, the regulation of CCL2 by CTNNAL1 is thought to be related to immune cells as well as the malignancy of cancer cells. Therefore, we will study the relationship between CTNNAL1 and CCL2 and their interaction with immune cells. To date, various signaling mechanisms involved in the secretion of CCL2 are unknown. Most researchers have reported that CCL2 is regulated by MAPK and NF-κB signaling, and mTORC1-FOXK1 signaling is also known [31,32,33]. These signals are all known to be important in CSCs.

This study suggests that CTNNAL1 may be a potential therapeutic target for the treatment of lung cancer and GBM.

## Figures and Tables

**Figure 1 biomedicines-11-01462-f001:**
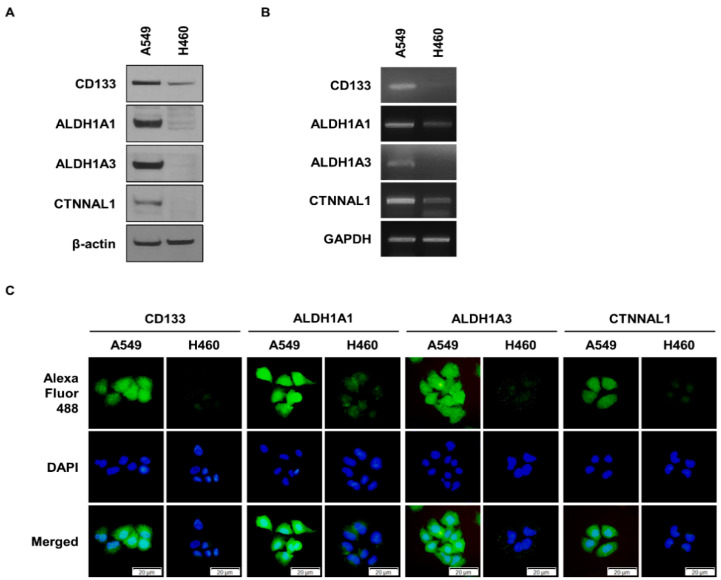
**Comparative analysis of CSC marker proteins and CTNNAL1 expression in A549 and H460.** (**A**) Comparison of protein expression levels of CSC marker proteins and CTNNAL1 between A549 and H460 cells. β-actin was used as a loading control. (**B**) Comparison of gene expression levels of CSC markers and CTNNAL1 between A549 cells and H460 cells. GAPDH was used as a loading control. (**C**) Comparison of protein expression levels of CSC marker proteins and CTNNAL1 using the ICC assay.

**Figure 2 biomedicines-11-01462-f002:**
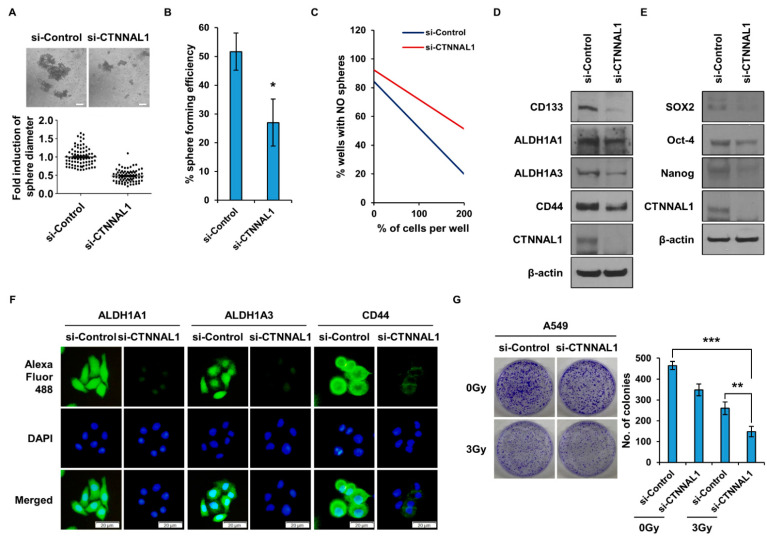
**Investigation of the role of CTNNAL1 in cancer stem cell properties and radio resistance.** (**A**) A549 cells were treated with si-CTNNAL1 and then exposed to a spheroidization assay using CM. (**B**) A549 cells were treated with si-CTNNAL1 and then subjected to a single-cell assay using a 96-well plate. Each well was seeded with a single cell and then cultured. (**C**) A549 cells were transfected with siRNA to suppress the expression of CTNNAL1 and then subjected to a limiting-dilution assay. Experiments were performed with 1, 50, 100, 150, and 200 cells in each well. (**D**) Expression of CSC marker proteins CD133, ALDH1A1, ALDH1A3, and CD44 after inhibition of CTNNAL1 using siRNA. (**E**) Comparison of the expression of CSC regulatory proteins SOX2, Oct-4, and Nanog after using si-CTNNAL1. (**F**) ICC analysis after knockdown of CTNNAL1 with siRNA to compare the expression of CSC marker proteins. (**G**) A colony-formation assay was performed to determine the ability to resist irradiation. The irradiation dose was 3 Gy, and the expression of CTNNAL1 was suppressed using siRNA before irradiation. After 10 days of culture, colonies were stained with crystal violet. Error bars represent mean ± SD of triplicate samples. * *p* < 0.05, ** *p* < 0.01, *** *p* < 0.0001 versus control. Scale bar = 50 μm.

**Figure 3 biomedicines-11-01462-f003:**
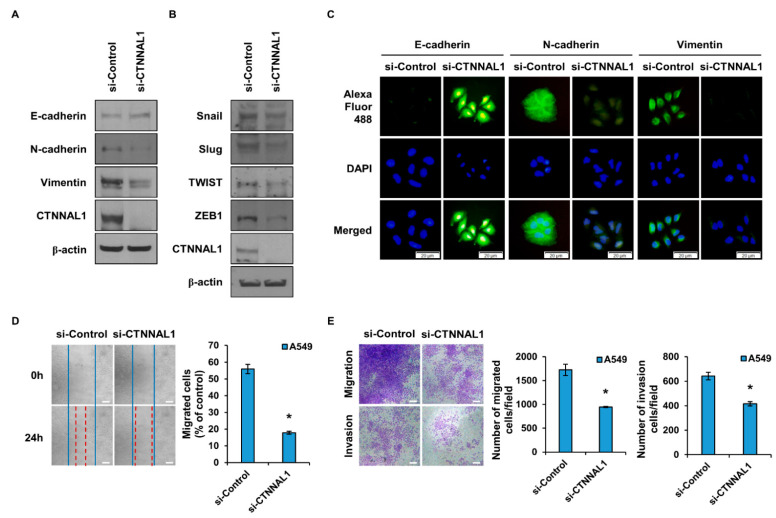
**Regulation of EMT and cell motility in A549 by CTNNAL1.** (**A**) Comparison of the expression of EMT marker proteins E-cadherin, N-cadherin, and Vimentin upon knockdown of CTNNAL1 using siRNA. (**B**) Comparison of the expression of EMT regulatory proteins Snail, Slug, TWIST, and ZEB1 in cells after si-CTNNAL1 treatment with si-Control. (**C**) An ICC assay was used to determine the expression of EMT marker proteins. siRNA was used to knock down the expression of CTNNAL1 in A549 cells. (**D**) A wound-healing assay was performed after treatment with siRNA to knock down the expression of CTNNAL1. (**E**) A migration and invasion assay was performed to analyze the migration and invasion abilities of cells using a Boyden chamber. Performed after A549 cells were treated with si-CTNNAL1 to suppress the gene. Error bars represent the mean ± SD of triplicate samples. * *p* < 0.001 versus control. Scale bar = 50 μm.

**Figure 4 biomedicines-11-01462-f004:**
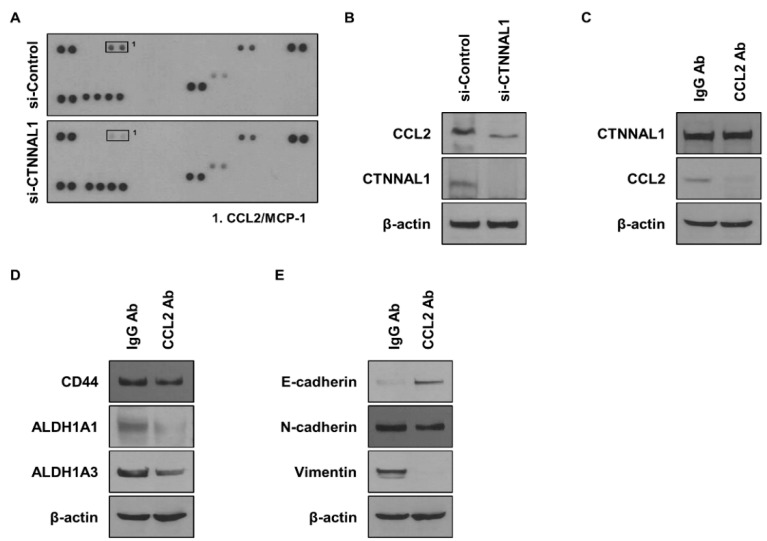
**Effects of CCL2 regulated by CTNNAL1 on lung cancer cells.** (**A**) Cytokine analysis using cytokine array. A549 cells were transfected with siRNA to knock down the CTNNAL1 gene. (**B**) Determination of CCL2 expression at the gene level. The expression of CTNNAL1 was knocked down using siRNA in A549 cells. (**C**) After treatment with CCL2 neutralizing antibody, the expression of CTNNAL1 was confirmed. (**D**) After treatment with CCL2 neutralizing antibodies, the expression of CSC marker proteins CD44, ALDH1A1, and ALDH1A3 was confirmed by WB. (**E**) Comparative analysis of EMT marker proteins E-cadherin, N-cadherin, and Vimentin after treatment with CCL2 neutralizing antibody.

**Figure 5 biomedicines-11-01462-f005:**
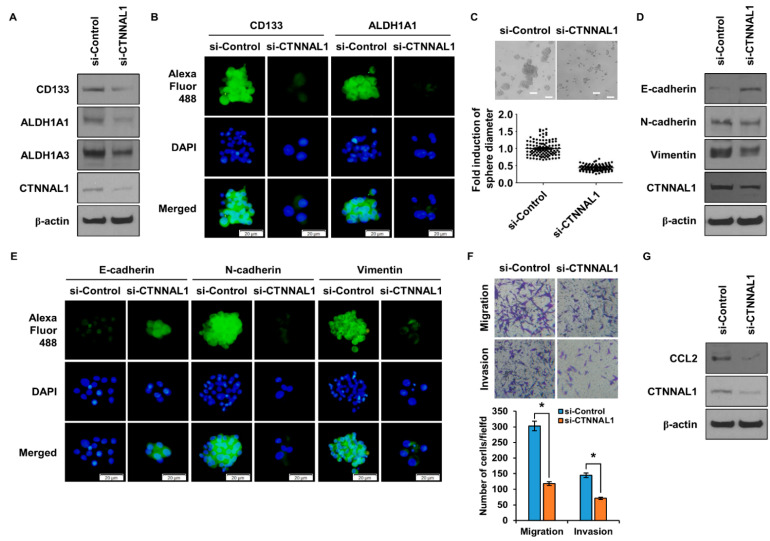
**Effect of CTNNAL1 on GSC and EMT markers in U87 cells.** (**A**) Marker proteins of GSCs, CD133, ALDH1A1, and ALDH1A3, were analyzed after CM treatment of U87 cells with CTNNAL1 knockdown by siRNA. (**B**) Marker proteins of GSCs were identified by an ICC analysis after CTNNAL1 gene silencing. Experiments were performed with treatment with CM. (**C**) A spheroidization assay was performed under CM conditions after knockdown of CTNNAL1 with siRNA in U87 cells. (**D**) Expression of EMT marker proteins E-cadherin, N-cadherin, and Vimentin was identified after silencing gene expression by treating U87 cells with si-CTNNAL1. (**E**) An ICC analysis was performed for visual analysis. The expression of marker proteins was analyzed after U87 cells were treated with si-CTNNAL1. (**F**) The migration and invasion abilities of U87 cells were analyzed after inhibiting the expression of CTNNAL1 by treatment with siRNA. (**G**) To determine whether CCL2 is regulated by CTNNAL1 in U87 cells, the expression of CCL2 was analyzed after inhibiting the expression of CTNNAL1 with siRNA. Error bars represent mean ± SD of triplicate samples. * *p* < 0.001 versus control. Scale bar = 50 μm.

## Data Availability

Not applicable.

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
