# Peer review of "Regulation of Cancer Stem Cells and Epithelial-Mesenchymal Transition by CTNNAL1 in Lung Cancer and Glioblastoma"

_biomedicines, 2023, doi:10.3390/biomedicines11051462_

Round 1
Reviewer 1 Report
This manuscript shows the function of CTNNAL1 int the lung cancer and glioblastoma cell lines. CTNNAL1 is correlated to cancer stemness and malignancy phenotype. I think the focused gene is interesting, however, there are several concerns.
Comments
Major
1. In Figure 3-5, A549 cells and U87 cells expressed both of epithelial marker genes and mesenchymal marker genes. A549 cells are epithelial-like cancer cells originated from lung epithelia. U87 cells also originated from glia or neuron-like cells. Why do the cell lines which cultured in the authors' condition express the both epithelial marker and mesenchymal marker. It is a very strange.
In addition, the authors showed the cell morphology with knockdown of CTNNAL1. The reviewer cannot see the EMT-like changes in these cell lines. The results related EMT and invasion phenotype are doubted.
2. Their experiments are depended on A549 and U87 cells. To demonstrate the hypothesis, the authors should perform the same experiment using other several cell lines.
3. What is the upstream regulation of CCL2? The connection between CCL2 and CTNAL1 is unclear. The authors should perform further evaluation and discussion.
Minor
4. What is shown in Figure 2C? The measurement method, calculation strategy, and statistical analysis are lacked. What does the figure demonsntarate?
5. There are many typos in the manuscript.
There are many typos in the manuscript. I recommend the English proofing.
Author Response
Dear. Reviewer
We sincerely thank you for your interest in our paper and your favorable review. We have answered the questions raised by the reviewer. We have done our best to answer the reviewer's questions and have filled in the gaps with additional experiments. We thank you again for taking the time to review our paper.
Best Regards,
Ph.D. Raekwon Kim

Reviewer 2 Report
Thank you for the opportunity to review this manuscript.
The authors have clearly demonstrated that CTNNAL1 regulated the expression of cancer stem cell markers and the ability of epithelial-mesenchymal transition (EMT) in lung cancer cells and glioma cells. Furthermore, they discovered that CTNNAL1 enhanced CCL2 expression, which is a new finding presented in the result section. I believe that most readers will find these results interesting.
However, I have a minor concern before publishing this in our journal.
The authors summarize the study in the last part of the introduction section (lines 58-61). I recommend moving the summary section. In the introduction, the authors also discuss the newly obtained CCL2, which should better be rewritten to provide a more comprehensive overview of the study's background, purpose, and significance.
Author Response
Dear. Reviewer
We sincerely thank you for your interest in our paper and your favorable review. We have answered the questions raised by the reviewer. We thank you again for taking the time to review our paper.
Best Regards,
Ph.D. Raekwon Kim

Round 2
Reviewer 1 Report
The authors have responded to the comments, although that is not fully satisfied by the reviewer.
English proofing is needed.